# Estimating LASSO Risk and Noise Level

**Mohsen Bayati**
Stanford University
bayati@stanford.edu

**Murat A. Erdogdu**
Stanford University
erdogdu@stanford.edu

**Andrea Montanari**
Stanford University
montanar@stanford.edu

## Abstract

We study the fundamental problems of variance and risk estimation in high dimensional statistical modeling. In particular, we consider the problem of learning a coefficient vector $\theta_0 \in \mathbb{R}^p$ from noisy linear observations $y = X\theta_0 + w \in \mathbb{R}^n$ ($p > n$) and the popular estimation procedure of solving the $\ell_1$-penalized least squares objective known as the LASSO or Basis Pursuit DeNoising (BPDN). In this context, we develop new estimators for the $\ell_2$ estimation risk $\|\widehat{\theta} - \theta_0\|_2$ and the variance of the noise when distributions of $\theta_0$ and $w$ are unknown. These can be used to select the regularization parameter optimally. Our approach combines Stein's unbiased risk estimate [Ste81] and the recent results of [BM12a][BM12b] on the analysis of approximate message passing and the risk of LASSO.

We establish high-dimensional consistency of our estimators for sequences of matrices $X$ of increasing dimensions, with independent Gaussian entries. We establish validity for a broader class of Gaussian designs, conditional on a certain conjecture from statistical physics.

To the best of our knowledge, this result is the first that provides an asymptotically consistent risk estimator for the LASSO solely based on data. In addition, we demonstrate through simulations that our variance estimation outperforms several existing methods in the literature.

## 1 Introduction

In Gaussian random design model for the linear regression, we seek to reconstruct an unknown coefficient vector $\theta_0 \in \mathbb{R}^p$ from a vector of noisy linear measurements $y \in \mathbb{R}^n$:

$$y = X\theta_0 + w, \tag{1.1}$$

where $X \in \mathbb{R}^{n \times p}$ is a measurement (or feature) matrix with iid rows generated through a multivariate normal density. The noise vector, $w$, has iid entries with mean $0$ and variance $\sigma^2$. While this problem is well understood in the low dimensional regime $p \ll n$, a growing corpus of research addresses the more challenging high-dimensional scenario in which $p > n$. The Basis Pursuit Denoising (BPDN) or LASSO [CD95, Tib96] is an extremely popular approach in this regime, that finds an estimate for $\theta_0$ by minimizing the following cost function

$$\mathcal{C}_{X,y}(\lambda, \theta) \equiv (2n)^{-1} \|y - X\theta\|_2^2 + \lambda\|\theta\|_1, \tag{1.2}$$

with $\lambda > 0$. In particular, $\theta_0$ is estimated by $\widehat{\theta}(\lambda; X, y) = \operatorname{argmin}_\theta \mathcal{C}_{X,y}(\lambda, \theta)$. This method is well suited for the ubiquitous case in which $\theta_0$ is sparse, i.e. a small number of features effectively predict the outcome. Since this optimization problem is convex, it can be solved efficiently, and fast specialized algorithms have been developed for this purpose [BT09].

Research has established a number of important properties of LASSO estimator under suitable conditions on the design matrix $X$, and for sufficiently sparse vectors $\theta_0$. Under irrepresentability conditions, the LASSO correctly recovers the support of $\theta_0$ [ZY06, MB06, Wai09]. Under weaker

conditions such as restricted isometry or compatibility properties the correct recovery of support fails however, the $\ell_2$ estimation error $\|\widehat{\theta} - \theta_0\|_2$ is of the same order as the one achieved by an oracle estimator that knows the support [CRT06, CT07, BRT09, BdG11]. Finally, [DMM09, RFG09, BM12b] provided asymptotic formulas for MSE or other operating characteristics of $\widehat{\theta}$, for Gaussian design matrices $X$.

While the aforementioned research provides solid justification for using the LASSO estimator, it is of limited guidance to the practitioner. For instance, a crucial question is how to set the regularization parameter $\lambda$. This question becomes even more urgent for high-dimensional methods with multiple regularization terms. The oracle bounds of [CRT06, CT07, BRT09, BdG11] suggest to take $\lambda = c\,\sigma\sqrt{\log p}$ with $c$ a dimension-independent constant (say $c = 1$ or $2$). However, in practice a factor two in $\lambda$ can make a substantial difference for statistical applications. Related to this issue is the question of estimating accurately the $\ell_2$ error $\|\widehat{\theta} - \theta_0\|_2^2$. The above oracle bounds have the form $\|\widehat{\theta} - \theta_0\|_2^2 \le C\,k\lambda^2$, with $k = \|\theta_0\|_0$ the number of nonzero entries in $\theta_0$, as long as $\lambda \ge c\sigma\sqrt{\log p}$. As a consequence, minimizing the bound does not yield a recipe for setting $\lambda$. Finally, estimating the noise level is necessary for applying these formulae, and this is in itself a challenging question.

The results of [DMM09, BM12b] provide exact asymptotic formulae for the risk, and its dependence on the regularization parameter $\lambda$. This might appear promising for choosing the optimal value of $\lambda$, but has one serious drawback. The formulae of [DMM09, BM12b] depend on the empirical distribution[1] of the entries of $\theta_0$, which is of course unknown, as well as on the noise level[2]. A step towards the resolution of this problem was taken in [DMM11], which determined the least favorable noise level and distribution of entries, and hence suggested a prescription for $\lambda$, and a predicted risk in this case. While this settles the question (in an asymptotic sense) from a minimax point of view, it would be preferable to have a prescription that is adaptive to the distribution of the entries of $\theta_0$ and to the noise level.

Our starting point is the asymptotic results of [DMM09, DMM11, BM12a, BM12b]. These provide a construction of an unbiased *pseudo-data* $\widehat{\theta}^u$ that is asymptotically Gaussian with mean $\theta_0$. The LASSO estimator $\widehat{\theta}$ is obtained by applying a denoiser function to $\widehat{\theta}^u$. We then use Stein's Unbiased Risk Estimate (*SURE*) [Ste81] to derive an expression for the $\ell_2$ risk (mean squared error) of this operation. What results is an expression for the mean squared error of the LASSO that only depends on the observed data $y$ and $X$. Finally, by modifying this formula we obtain an estimator for the noise level.

We prove that these estimators are asymptotically consistent for sequences of design matrices $X$ with converging aspect ratio and iid Gaussian entries. We expect that the consistency holds far beyond this case. In particular, for the case of general Gaussian design matrices, consistency holds conditionally on a conjectured formula stated in [JM13] on the basis of the "replica method" from statistical physics.

For the sake of concreteness, let us briefly describe our method in the case of standard Gaussian design that is when the design matrix $X$ has iid Gaussian entries. We construct the unbiased pseudo-data vector by

$$\widehat{\theta}^u = \widehat{\theta} + X^T(y - X\widehat{\theta})/[n - \|\widehat{\theta}\|_0]. \tag{1.3}$$

Our estimator of the mean squared error is derived from applying SURE to unbiased pseudo-data. In particular, our estimator is $\widehat{R}(y, X, \lambda, \widehat{\tau})$ where

$$\widehat{R}(y, X, \lambda, \tau) \equiv \tau^2 \left( 2\|\widehat{\theta}\|_0/p - 1 \right) + \|X^T(y - X\widehat{\theta})\|_2^2 \Big/ \left[ p(n - \|\widehat{\theta}\|_0)^2 \right] \tag{1.4}$$

Here $\widehat{\theta}(\lambda; X, y)$ is the LASSO estimator and $\widehat{\tau} = \|y - X\widehat{\theta}\|_2/[n - \|\widehat{\theta}\|_0]$.

Our estimator of the noise level is

$$\widehat{\sigma}^2/n = \widehat{\tau}^2 - \widehat{R}(y, X, \lambda, \widehat{\tau})/\delta$$

where $\delta = n/p$. Although our rigorous results are asymptotic in the problem dimensions, we show through numerical simulations that they are accurate already on problems with a few thousands of

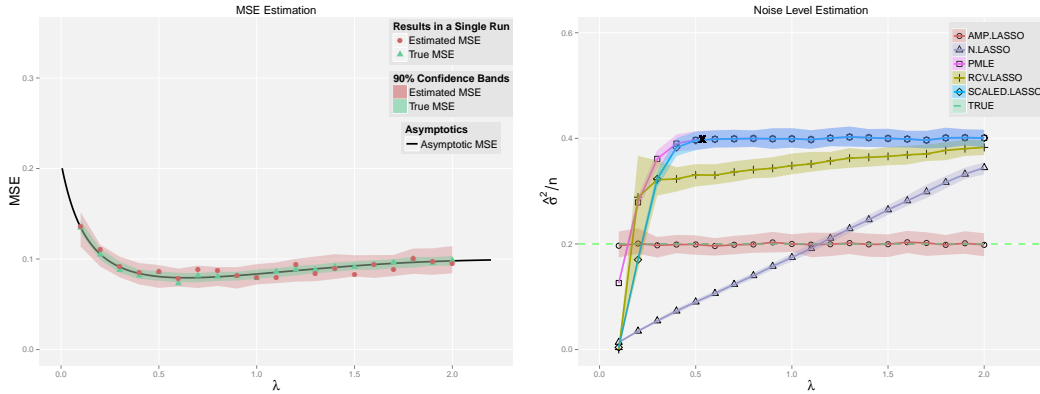

Figure 1: Red color represents the estimated values by our estimators and green color represents the true values to be estimated. *Left:* MSE versus regularization parameter $\lambda$. Here, $\delta = 0.5$, $\sigma^2/n = 0.2$, $X \in \mathbb{R}^{n \times p}$ with iid $\mathsf{N}_1(0,1)$ entries where $n = 4000$. *Right:* $\hat{\sigma}^2/n$ versus $\lambda$. Comparison of different estimators of $\sigma^2$ under the same model parameters. Scaled Lasso's prescribed choice of $(\lambda, \hat{\sigma}^2/n)$ is marked with a bold **x**.

variables. To the best of our knowledge, this is the first method for estimating the LASSO mean square error solely based on data. We compare our approach with earlier work on the estimation of the noise level. The authors of [NSvdG10] target this problem by using a $\ell_1$-penalized maximum log-likelihood estimator (PMLE) and a related method called "Scaled Lasso" [SZ12] (also studied by [BC13]) considers an iterative algorithm to jointly estimate the noise level and $\theta_0$. Moreover, authors of [FGH12] developed a refitted cross-validation (RCV) procedure for the same task. Under some conditions, the aforementioned studies provide consistency results for their noise level estimators. We compare our estimator with these methods through extensive numerical simulations.

The rest of the paper is organized as follows. In order to motivate our theoretical work, we start with numerical simulations in Section 2. The necessary background on SURE and asymptotic distributional characterization of the LASSO is presented in Section 3. Finally, our main theoretical results can be found in Section 4.

## 2 Simulation Results

In this section, we validate the accuracy of our estimators through numerical simulations. We also analyze the behavior of our variance estimator as $\lambda$ varies, along with four other methods. Two of these methods rely on the minimization problem,

$$(\widehat{\theta}, \widehat{\sigma}) = \mathrm{argmin}_{\theta, \sigma} \left\{ \frac{\|y - X\theta\|_2^2}{2nh_1(\sigma)} + h_2(\sigma) + \lambda \, \frac{\|\theta\|_1}{2^3 \, h_3(\sigma)} \right\},$$

where for PMLE $h_1(\sigma) = \sigma^2$, $h_2(\sigma) = \log(\sigma)$, $h_3(\sigma) = \sigma$ and for the Scaled Lasso $h_1(\sigma) = \sigma$, $h_2(\sigma) = \sigma/2$, and $h_3(\sigma) = 1$. The third method is a *naïve* procedure that estimates the variance in two steps: (i) use the LASSO to determine the relevant variables; (ii) apply ordinary least squares on the selected variables to estimate the variance. The fourth method is *Refitted Cross-Validation* (RCV) by [FGH12] which also has two-stages. RCV requires *sure screening property* that is the model selected in its first stage includes all the relevant variables. Note that this requirement may not be satisfied for many values of $\lambda$. In our implementation of RCV, we used the LASSO for variable selection.

In our simulation studies, we used the LASSO solver `l1_ls` [SJKG07]. We simulated across 50 replications within each, we generated a new Gaussian design matrix $X$. We solved for LASSO over 20 equidistant $\lambda$'s in the interval $[0.1, 2]$. For each $\lambda$, a new signal $\theta_0$ and noise independent from $X$ were generated.

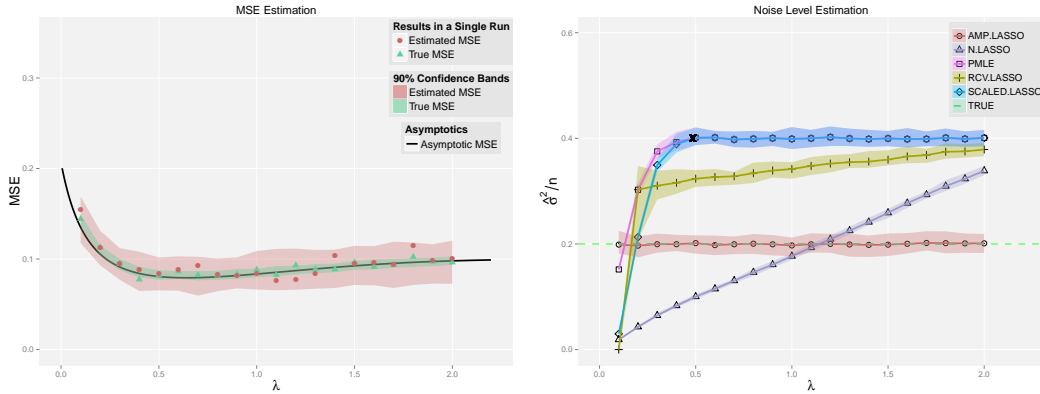

Figure 2: Red color represents the estimated values by our estimators and green color represents the true values to be estimated. *Left:* MSE versus regularization parameter $\lambda$. Here, $\delta = 0.5$, $\sigma^2/n = 0.2$, rows of $X \in \mathbb{R}^{n \times p}$ are iid from $\mathsf{N}_p(0, \Sigma)$ where $n = 5000$ and $\Sigma$ has entries 1 on the main diagonal, 0.4 on above and below the main diagonal. *Right:* Comparison of different estimators of $\sigma^2/n$. Parameter values are the same as in Figure 1. Scaled Lasso's prescribed choice of $(\lambda, \hat{\sigma}^2/n)$ is marked with a bold **x**.

The results are demonstrated in Figures 1 and 2. Figure 1 is obtained using $n = 4000$, $\delta = 0.5$ and $\sigma^2/n = 0.2$. The coordinates of true signal independently get values 0, 1, $-1$ with probabilities 0.9, 0.05, 0.05 respectively. For each replication, we used a design matrix $X$ where $X_{i,j} \overset{iid}{\sim} \mathsf{N}_1(0, 1)$. Figure 2 is obtained with $n = 5000$ and same values of $\delta$ and $\sigma^2$ as in Figure 1. The coordinates of true signal independently get values 0, 1, $-1$ with probabilities 0.9, 0.05, 0.05 respectively. For each replication, we used a design matrix $X$ where each row is independently generated through $\mathsf{N}_p(0, \Sigma)$ where $\Sigma$ has 1 on the main diagonal and 0.4 above and below the diagonal.

As can be seen from the figures, the asymptotic theory applies quite well to the finite dimensional data. We refer reader to [BEM13] for a more detailed simulation analysis.

## 3 Background and Notations

### 3.1 Preliminaries and Definitions

First, we need to provide a brief introduction to *approximate message passing* (AMP) algorithm suggested by [DMM09] and its connection to LASSO (see [DMM09, BM12b] for more details).

For an appropriate sequence of non-linear denoisers $\{\eta_t\}_{t \geq 0}$, the AMP algorithm constructs a sequence of estimates $\{\theta^t\}_{t \geq 0}$, pseudo-data $\{y^t\}_{t \geq 0}$, and residuals $\{\epsilon^t\}_{t \geq 0}$ where $\theta^t, y_t \in \mathbb{R}^p$ and $\epsilon^t \in \mathbb{R}^n$. These sequences are generated according to the iteration

$$\theta^{t+1} = \eta_t(y^t), \quad y^t = \theta^t + X^T \epsilon^t/n, \quad \epsilon^t = y - X\theta^t + \frac{1}{\delta}\epsilon^{t-1} \langle \eta_t'(y^{t-1}) \rangle, \qquad (3.1)$$

where $\delta \equiv n/p$ and the algorithm is initialized with $\theta^0 = \epsilon^0 = 0 \in \mathbb{R}^p$. In addition, each denoiser $\eta_t(\cdot)$ is a separable function and its derivative is denoted by $\eta_t'(\cdot)$. Given a scalar function $f$ and a vector $u \in \mathbb{R}^m$, we let $f(u)$ denote the vector $(f(u_1), \ldots, f(u_m)) \in \mathbb{R}^m$ obtained by applying $f$ component-wise and $\langle u \rangle \equiv m^{-1} \sum_{i=1}^m u_i$ is the average of the vector $u \in \mathbb{R}^m$.

Next, consider the *state evolution* for the AMP algorithm. For the random variable $\Theta_0 \sim p_{\theta_0}$, a positive constant $\sigma^2$ and a given sequence of non-linear denoisers $\{\eta_t\}_{t \geq 0}$, define the sequence $\{\tau_t^2\}_{t \geq 0}$ iteratively by

$$\tau_{t+1}^2 = \mathsf{F}_t(\tau_t^2), \qquad \mathsf{F}_t(\tau^2) \equiv \sigma^2 + \frac{1}{\delta} \mathbb{E}\{ [\eta_t(\Theta_0 + \tau Z) - \Theta_0]^2 \}, \qquad (3.2)$$

where $\tau_0^2 = \sigma^2 + \mathbb{E}\{\Theta_0^2\}/\delta$ and $Z \sim \mathsf{N}_1(0, 1)$ is independent of $\Theta_0$. From Eq. 3.2, it is apparent that the function $\mathsf{F}_t$ depends on the distribution of $\Theta_0$. It is shown in [BM12a] that the pseudo-data

$y^t$ has the same asymptotic distribution as $\Theta_0 + \tau_t Z$. This result can be roughly interpreted as the pseudo-data generated by AMP is the summation of the true signal and a normally distributed noise which has zero mean. Its variance is determined by the state evolution. In other words, each iteration produces a pseudo-data that is distributed normally around the true signal, i.e. $y_i^t \approx \theta_{0,i} + \mathsf{N}_1(0, \tau_t^2)$. The importance of this result will appear later when we use Stein's method in order to obtain an estimator for the MSE and the variance of the noise.

We will use state evolution in order to describe the behavior of a specific type of converging sequence defined as the following:

**Definition 1.** *The sequence of instances $\{\theta_0(n), X(n), \sigma^2(n)\}_{n \in \mathbb{N}}$ indexed by $n$ is said to be a converging sequence if $\theta_0(n) \in \mathbb{R}^p$, $X(n) \in \mathbb{R}^{n \times p}$, $\sigma^2(n) \in \mathbb{R}$ and $p = p(n)$ is such that $n/p \to \delta \in (0, \infty)$, $\sigma^2(n)/n \to \sigma_0^2$ for some $\sigma_0 \in \mathbb{R}$ and in addition the following conditions hold:*

*(a) The empirical distribution of $\{\theta_{0,i}(n)\}_{i=1}^p$, converges in distribution to a probability measure $p_{\theta_0}$ on $\mathbb{R}$ with bounded $2^{nd}$ moment. Further, as $n \to \infty$, $p^{-1} \sum_{i=1}^p \theta_{0,i}(n)^2 \to \mathbb{E}_{p_{\theta_0}}\{\Theta_0^2\}$.*

*(b) If $\{e_i\}_{1 \leq i \leq p} \subset \mathbb{R}^p$ denotes the standard basis, then $n^{-1/2} \max_{i \in [p]} \|X(n)e_i\|_2 \to 1$, $n^{-1/2} \min_{i \in [p]} \|X(n)e_i\|_2 \to 1$, as $n \to \infty$ with $[p] \equiv \{1, \ldots, p\}$.*

We provide rigorous results for the special class of converging sequences when entries of $X$ are iid $\mathsf{N}_1(0,1)$ (i.e., *standard gaussian design model*). We also provide results (assuming Conjecture 4.4 is correct) when rows of $X$ are iid multivariate normal $\mathsf{N}_p(0, \Sigma)$ (i.e., *general gaussian design model*).

In order to discuss the LASSO connection for the AMP algorithm, we need to use a specific class of denoisers and apply an appropriate calibration to the state evolution. Here, we provide briefly how this can be done and we refer the reader to [BEM13] for a detailed discussion.

Denote by $\eta : \mathbb{R} \times \mathbb{R}_+ \to \mathbb{R}$ the soft thresholding denoiser where

$$\eta(x; \xi) = \left\{ \begin{array}{ll} x - \xi & \text{if } x > \xi \\ 0 & \text{if } -\xi \leq x \leq \xi \\ x + \xi & \text{if } x < -\xi \end{array} \right. .$$

Also, denote by $\eta'(\,\cdot\,;\,\cdot\,)$, the derivative of the soft-thresholding function with respect to its first argument. We will use the AMP algorithm with the soft-thresholding denoiser $\eta_t(\,\cdot\,) = \eta(\,\cdot\,;\xi_t)$ along with a suitable sequence of thresholds $\{\xi_t\}_{t \geq 0}$ in order to obtain a connection to the LASSO.

Let $\alpha > 0$ be a constant and at every iteration $t$, choose the threshold $\xi_t = \alpha\tau_t$. It was shown in [DMM09] and [BM12b] that the state evolution has a unique fixed point $\tau_* = \lim_{t \to \infty} \tau_t$, and there exists a mapping $\alpha \mapsto \tau_*(\alpha)$, between those two parameters. Further, it was shown that a function $\alpha \mapsto \lambda(\alpha)$ with domain $(\alpha_{\min}(\delta), \infty)$ for some constant $\alpha_{\min}$, and given by

$$\lambda(\alpha) \equiv \alpha\tau_* \big(1 - \frac{1}{\delta}\mathbb{E}\big[\eta'(\Theta_0 + \tau_* Z; \alpha\tau_*)\big]\big),$$

admits a well-defined continuous and non-decreasing inverse $\alpha : (0, \infty) \to (\alpha_{\min}, \infty)$. In particular, the functions $\lambda \mapsto \alpha(\lambda)$ and $\alpha \mapsto \tau_*(\alpha)$ provide a calibration between the AMP algorithm and the LASSO where $\lambda$ is the regularization parameter.

### 3.2 Distributional Results for the LASSO

We will proceed by stating a distributional result on LASSO which was established in [BM12b].

**Theorem 3.1.** *Let $\{\theta_0(n), X(n), \sigma^2(n)\}_{n \in \mathbb{N}}$ be a converging sequence of instances of the standard Gaussian design model. Denote the LASSO estimator of $\theta_0(n)$ by $\widehat{\theta}(n, \lambda)$ and the unbiased pseudo-data generated by LASSO by $\widehat{\theta}^u(n, \lambda) \equiv \widehat{\theta} + X^T(y - X\widehat{\theta})/[n - \|\widehat{\theta}\|_0]$.*

*Then, as $n \to \infty$, the empirical distribution of $\{\widehat{\theta}_i^u, \theta_{0,i}\}_{i=1}^p$ converges weakly to the joint distribution of $(\Theta_0 + \tau_* Z, \Theta_0)$ where $\Theta_0 \sim p_{\theta_0}$, $\tau_* = \tau_*(\alpha(\lambda))$, $Z \sim \mathsf{N}_1(0,1)$ and $\Theta_0$ and $Z$ are independent random variables.*

The above theorem combined with the stationarity condition of the LASSO implies that the empirical distribution of $\{\widehat{\theta}_i, \theta_{0,i}\}_{i=1}^p$ converges weakly to the joint distribution of $\big(\eta(\Theta_0 + \tau_* Z; \xi_*), \Theta_0\big)$

where $\xi_* = \alpha(\lambda)\tau_*(\alpha(\lambda))$. It is also important to emphasize a relation between the asymptotic MSE, $\tau_*^2$ and the model variance. By Theorem 3.1 and the state evolution recursion, almost surely,

$$\lim_{p\to\infty} \|\widehat{\theta} - \theta_0\|_2^2/p = \mathbb{E}\left[\left[\eta(\Theta_0 + \tau_* Z; \xi_*) - \Theta_0\right]^2\right] = \delta(\tau_*^2 - \sigma_0^2), \qquad (3.3)$$

which will be helpful to get an estimator for the noise level.

## 3.3 Stein's Unbiased Risk Estimator

In [Ste81], Stein proposed a method to estimate the risk of an almost arbitrary estimator of the mean of a multivariate normal vector. A generalized form of his method can be stated as the following.

**Proposition 3.2.** *[Ste81]&[Joh12] Let $x, \mu \in \mathbb{R}^n$ and $\boldsymbol{V} \in \mathbb{R}^{n\times n}$ be such that $x \sim \mathsf{N}_n(\mu, \boldsymbol{V})$. Suppose that $\hat{\mu}(x) \in \mathbb{R}^n$ is an estimator of $\mu$ for which $\hat{\mu}(x) = x + g(x)$ and that $g : \mathbb{R}^n \to \mathbb{R}^n$ is weakly differentiable and that $\forall i, j \in [n]$, $\mathbb{E}_\nu[|x_i g_i(x)| + |x_j g_j(x)|] < \infty$ where $\nu$ is the measure corresponding to the multivariate Gaussian distribution $\mathsf{N}_n(\mu, \boldsymbol{V})$. Define the functional*

$$S(x, \hat{\mu}) \equiv \mathrm{Tr}(\boldsymbol{V}) + 2\mathrm{Tr}(\boldsymbol{V} Dg(x)) + \|g(x)\|_2^2,$$

*where $Dg$ is the vector derivative. $S(x, \hat{\mu})$ is an unbiased estimator of the risk, i.e. $\mathbb{E}_\nu\|\hat{\mu}(x) - \mu\|_2^2 = \mathbb{E}_\nu[S(x, \hat{\mu})]$.*

In the literature of statistics, the above estimator is called "Stein's Unbiased Risk Estimator" or SURE. The following remark will be helpful to build intuition about our approach.

**Remark 1.** *If we consider the risk of soft thresholding estimator $\eta(x_i; \xi)$ for $\mu_i$ when $x_i \sim \mathsf{N}_1(\mu_i, \sigma^2)$ for $i \in [m]$, the above formula suggests the functional*

$$\frac{S(x, \eta(\,\cdot\,; \xi))}{m} = \sigma^2 - \frac{2\sigma^2}{m}\sum_{i=1}^{m} \mathbb{1}_{\{|x_i| \le \xi\}} + \frac{1}{m}\sum_{i=1}^{m}\left[\min\{|x_i|, \xi\}\right]^2,$$

*as an estimator of the corresponding MSE.*

# 4 Main Results

## 4.1 Standard Gaussian Design Model

We start by defining two estimators that are motivated by Proposition 3.2.

**Definition 2.** *Define*

$$\widehat{R}_\psi(x, \tau) \equiv -\tau^2 + 2\tau^2\langle\psi'(x)\rangle + \langle(\psi(x) - x)^2\rangle,$$

*where $x \in \mathbb{R}^m$, $\tau \in \mathbb{R}_+$, and $\psi : \mathbb{R} \to \mathbb{R}$ is a suitable non-linear function. Also for $y \in \mathbb{R}^n$ and $X \in \mathbb{R}^{n\times p}$ denote by $\widehat{R}(y, X, \lambda, \tau)$, the estimator of the mean squared error of LASSO where*

$$\widehat{R}(y, X, \lambda, \tau) \equiv \frac{\tau^2}{p}(2\|\widehat{\theta}\|_0 - p) + \frac{\|X^T(y - X\widehat{\theta})\|_2^2}{p(n - \|\widehat{\theta}\|_0)^2}.$$

**Remark 2.** *Note that $\widehat{R}(y, X, \lambda, \tau)$ is just a special case of $\widehat{R}_\psi(x, \tau)$ when $x = \widehat{\theta}^u$ and $\psi(\,\cdot\,) = \eta(\,\cdot\,; \xi)$ for $\xi = \lambda/(1 - \|\widehat{\theta}\|_0/p)$.*

We are now ready to state the following theorem on the asymptotic MSE of the AMP:

**Theorem 4.1.** *Let $\{\theta_0(n), X(n), \sigma^2(n)\}_{n\in\mathbb{N}}$ be a converging sequence of instances of the standard Gaussian design model. Denote the sequence of estimators of $\theta_0(n)$ by $\{\theta^t(n)\}_{t\ge 0}$, the pseudo-data by $\{y^t(n)\}_{t\ge 0}$, and residuals by $\{\epsilon^t(n)\}_{t\ge 0}$ produced by AMP algorithm using the sequence of Lipschitz continuous functions $\{\eta_t\}_{t\ge 0}$ as in Eq. 3.1.*

*Then, as $n \to \infty$, the mean squared error of the AMP algorithm at iteration $t+1$ has the same limit as $\widehat{R}_{\eta_t}(y^t, \widehat{\tau})$ where $\widehat{\tau} = \|\epsilon^t\|_2/\sqrt{n}$. More precisely, with probability one,*

$$\lim_{n\to\infty} \|\theta^{t+1} - \theta_0\|_2^2/p(n) = \lim_{n\to\infty} \widehat{R}_{\eta_t}(y^t, \widehat{\tau}). \qquad (4.1)$$

*In other words, $\widehat{R}_{\eta_t}(y^t, \widehat{\tau})$ is a consistent estimator of the asymptotic mean squared error of the AMP algorithm at iteration $t+1$.*

The above theorem allows us to accurately predict how far the AMP estimate is from the true signal at iteration $t + 1$ and this can be utilized as a stopping rule for the AMP algorithm. Note that it was shown in [BM12b] that the left hand side of Eq. (4.1) is $\mathbb{E}[(\eta_t(\Theta_0 + \tau_t Z) - \Theta_0)^2]$. Combining this with the above theorem, we easily obtain,

$$\lim_{n \to \infty} \widehat{R}_{\eta_t}(y^t, \widehat{\tau}_t) = \mathbb{E}[(\eta_t(\Theta_0 + \tau_t Z) - \Theta_0)^2].$$

We state the following version of Theorem 4.1 for the LASSO.

**Theorem 4.2.** *Let $\{\theta_0(n), X(n), \sigma^2(n)\}_{n \in \mathbb{N}}$ be a converging sequence of instances of the standard Gaussian design model. Denote the LASSO estimator of $\theta_0(n)$ by $\widehat{\theta}(n, \lambda)$. Then with probability one,*

$$\lim_{n \to \infty} \|\widehat{\theta} - \theta_0\|_2^2 / p(n) = \lim_{n \to \infty} \widehat{R}(y, X, \lambda, \widehat{\tau}),$$

*where $\widehat{\tau} = \|y - X\widehat{\theta}\|_2 / [n - \|\widehat{\theta}\|_0]$. In other words, $\widehat{R}(y, X, \lambda, \widehat{\tau})$ is a consistent estimator of the asymptotic mean squared error of the LASSO.*

Note that Theorem 4.2 enables us to assess the quality of the LASSO estimation without knowing the true signal itself or the noise (or their distribution). The following corollary can be shown using the above theorem and Eq. 3.3.

**Corollary 4.3.** *In the standard Gaussian design model, the variance of the noise can be accurately estimated by $\widehat{\sigma}^2 / n \equiv \widehat{\tau}^2 - \widehat{R}(y, X, \lambda, \widehat{\tau}) / \delta$ where $\delta = n/p$ and other variables are defined as in Theorem 4.2. In other words, we have*

$$\lim_{n \to \infty} \hat{\sigma}^2 / n = \sigma_0^2, \tag{4.2}$$

*almost surely, providing us a consistent estimator for the variance of the noise in the LASSO.*

**Remark 3.** *Theorems 4.1 and 4.2 provide a rigorous method for selecting the regularization parameter optimally. Also, note that obtaining the expression in Theorem 4.2 only requires solving one solution path to LASSO problem versus $k$ solution paths required by $k$-fold cross-validation methods. Additionally, using the exponential convergence of AMP algorithm for the standard gaussian design model, proved by [BM12b], one can use $O(\log(1/\epsilon))$ iterations of AMP algorithm and Theorem 4.1 to obtain the solution path with an additional error up to $O(\epsilon)$.*

## 4.2 General Gaussian Design Model

In Section 4.1, we devised our estimators based on the standard Gaussian design model. Motivated by Theorem 4.2, we state the following conjecture of [JM13].

Let $\{\Omega(n)\}_{n \in \mathbb{N}}$ be a sequence of inverse covariance matrices. Define the general Gaussian design model by the converging sequence of instances $\{\theta_0(n), X(n), \sigma^2(n)\}_{n \in \mathbb{N}}$ where for each $n$, rows of design matrix $X(n)$ are *iid* multivariate Gaussian, i.e. $\mathsf{N}_p(0, \Omega(n)^{-1})$.

**Conjecture 4.4** ([JM13]). *Let $\{\theta_0(n), X(n), \sigma^2(n)\}_{n \in \mathbb{N}}$ be a converging sequence of instances under the general Gaussian design model with a sequence of proper inverse covariance matrices $\{\Omega(n)\}_{n \in \mathbb{N}}$. Assume that the empirical distribution of $\{(\theta_{0,i}, \Omega_{ii}\}_{i=1}^p$ converges weakly to the distribution of a random vector $(\Theta_0, \Upsilon)$. Denote the LASSO estimator of $\theta_0(n)$ by $\widehat{\theta}(n, \lambda)$ and the LASSO pseudo-data by $\widehat{\theta}^u(n, \lambda) \equiv \widehat{\theta} + \Omega X^T (y - X\widehat{\theta}) / [n - \|\widehat{\theta}\|_0]$. Then, for some $\tau \in \mathbb{R}$, the empirical distribution of $\{\theta_{0,i}, \widehat{\theta}_i^u, \Omega_{ii}\}$ converges weakly to the joint distribution of $(\Theta_0, \Theta_0 + \tau \Upsilon^{1/2} Z, \Upsilon)$, where $Z \sim \mathsf{N}_1(0, 1)$, and $(\Theta_0, \Upsilon)$ are independent random variables. Further, the empirical distribution of $(y - X\widehat{\theta}) / [n - \|\widehat{\theta}\|_0]$ converges weakly to $\mathsf{N}(0, \tau^2)$.*

A heuristic justification of this conjecture using the replica method from statistical physics is offered in [JM13]. Using the above conjecture, we define the following generalized estimator of the linearly transformed risk under the general Gaussian design model. The construction of the estimator is essentially the same as before i.e. apply SURE to unbiased pseudo-data.

**Definition 3.** *For an inverse covariance matrix $\Omega$ and a suitable matrix $V \in \mathbb{R}^{p \times p}$, let $W = V\Omega V^T$ and define an estimator of $\|V(\widehat{\theta} - \theta)\|_2^2/p$ as*

$$\widehat{\Gamma}_\Omega(y, X, \tau, \lambda, V) = \frac{\tau^2}{p}\left(\mathrm{Tr}\,(W_{SS}) - \mathrm{Tr}\,(W_{\tilde{S}\tilde{S}}) - 2\mathrm{Tr}\left(W_{\tilde{S}S}\Omega_{S\tilde{S}}\Omega_{\tilde{S}\tilde{S}}^{-1}\right)\right) + \frac{\|V\Omega X^T(y - X\widehat{\theta})\|_2^2}{p(n - \|\widehat{\theta}\|_0)^2}$$

*where $y \in \mathbb{R}^n$ and $X \in \mathbb{R}^{n \times p}$ denote the linear observations and the design matrix, respectively. Further, $\widehat{\theta}(n, \lambda)$ is the* LASSO *solution for penalty level $\lambda$ and $\tau$ is a real number. $S \subset [p]$ is the support of $\widehat{\theta}$ and $\tilde{S}$ is $[p] \setminus S$. Finally, for a $p \times p$ matrix $M$ and subsets $D, E$ of $[p]$ the notation $M_{DE}$ refers to the $|D| \times |E|$ sub-matrix of $M$ obtained by intersection of rows with indices from $D$ and columns with indices from $E$.*

Derivation of the above formula is rather complicated and we refer the reader to [BEM13] for a detailed argument. A notable case, when $V = I$, corresponds to the mean squared error of LASSO for the general Gaussian design and the estimator $\widehat{R}(y, X, \lambda, \tau)$ is just a special case of the estimator $\widehat{\Gamma}_\Omega(y, X, \tau, \lambda, V)$. That is, when $V = \Omega = I$, we have $\widehat{\Gamma}_I(y, X, \tau, \lambda, I) = \widehat{R}(y, X, \lambda, \tau)$.

Now, we state the following analog of Theorem 4.2.

**Theorem 4.5.** *Let $\{\theta_0(n), X(n), \sigma^2(n)\}_{n \in \mathbb{N}}$ be a converging sequence of instances of the general Gaussian design model with the inverse covariance matrices $\{\Omega(n)\}_{n \in \mathbb{N}}$. Denote the* LASSO *estimator of $\theta_0(n)$ by $\widehat{\theta}(n, \lambda)$. If Conjecture 4.4 holds, then, with probability one,*

$$\lim_{n \to \infty} \|\widehat{\theta} - \theta_0\|_2^2/p(n) = \lim_{n \to \infty} \widehat{\Gamma}_\Omega(y, X, \widehat{\tau}, \lambda, I)$$

*where $\widehat{\tau} = \|y - X\widehat{\theta}\|_2/[n - \|\widehat{\theta}\|_0]$. In other words, $\widehat{\Gamma}_\Omega(y, X, \widehat{\tau}, \lambda, I)$ is a consistent estimator of the asymptotic* MSE *of the* LASSO.

We will assume that a similar state evolution holds for the general design. In fact, for the general case, replica method suggests the relation

$$\lim_{n \to \infty} \|\Omega^{-\frac{1}{2}}(\widehat{\theta} - \theta)\|_2^2/p(n) = \delta(\tau^2 - \sigma_0^2).$$

Hence motivated by the Corollary 4.3, we state the following result on the general Gaussian design model.

**Corollary 4.6.** *Assume that Conjecture 4.4 holds. In the general Gaussian design model, the variance of the noise can be accurately estimated by*

$$\hat{\sigma}^2(n, \Omega)/n \equiv \widehat{\tau}^2 - \widehat{\Gamma}_\Omega(y, X, \widehat{\tau}, \lambda, \Omega^{-\frac{1}{2}})/\delta,$$

*where $\delta = n/p$ and other variables are defined as in Theorem 4.5. Also, we have*

$$\lim_{n \to \infty} \hat{\sigma}^2/n = \sigma_0^2,$$

*almost surely, providing us a consistent estimator for the noise level in* LASSO.

Corollary 4.6, extends the results stated in Corollary 4.3 to the general Gaussian design matrices. The derivation of formulas in Theorem 4.5 and Corollary 4.6 follows similar arguments as in the standard Gaussian design model. In particular, they are obtained by applying SURE to the distributional result of Conjecture 4.4 and using the stationary condition of the LASSO. Details of this derivation can be found in [BEM13].

## Footnotes

[1] The probability distribution that puts a point mass $1/p$ at each of the p entries of the vector.

[2] Note that our definition of noise level $\sigma$ corresponds to $\sigma\sqrt{n}$ in most of the compressed sensing literature.

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
