[Reviews · NeurIPS 2013]

Submitted by Assigned_Reviewer_4

This work proposes a new method for estimating the risk of the Lasso procedure,
providing also an estimate of the unknown noise level, in the Gaussian regression
model. The considerations are based on asymptotic results relying on
approximate message passing as well as on the Stein Lemma.

The paper offers asymptotic guarantees in the regime where n/p converges to
a certain level. I also contain a short experiment section.

The general impression is that the paper is not totally finished (cf. the amount
of typos below), and is a large collection of difficult results that are not
necessarily explained in details.
Moreover the experiment section is not really convincing, since the comparisons
given are only considering naive procedure (or so).


It is also to be noted that all the results page 8 rely on a conjecture,
and therefore cannot be taken for granted as written.


Recent works have also focused on the Lasso estimator when the noise level is unknown,
and several propositions are worth mentioning. Among others:

"L1-Penalization for Mixture Regression Models",
Nicolas Städler, Peter Bühlmann, Sara van de Geer, 2010

"Pivotal Estimation of Nonparametric Functions via Square-root Lasso"
Alexandre Belloni, Victor Chernozhukov, Lie Wang, 2011

"SOCP based variance free Dantzig Selector with application to robust estimation"
Arnak S. Dalalyan, 2012


Comparisons with some of those methods is highly needed to validate in practice
the results obtained.




Questions: can the authors relax the assumption that n/p converges to a fixed ratio?
what about cases where p > > n?

comments:

-l83: "for of "- > "for"

-l100: latex issue...

-l153: lambda-> latex lambda
Moreover I do not understand why when changing the lambda, the signal is re-sample
again. Is there any particular reason for that?

-l157: delta is defined only afterwards. Please correct.

-l179: tha?

-l190: no clue is given for choosing y_0...

-l208/214: the N_1 is different in the 2 lines, please correct the typo.

-l295 and 300: what is nu?

-l297: what is D?

-l307: it seems that the normalization differs in that line w.r.t. the equation line 297.

-l442: this book simply does not exist. The one I know is co-written by S. Van de Geer. Please
correct.

-l445/450: Candes should be Cand\`es.

-In the references harmonize the authors name so that they always appear with the same spelling.
e. g. A. Montanari is sometimes also Andrea Montari.

-l472 is corrupted.

-l498: define the scalar product used with the 1/p normalization.

-l513: lemma lemma... lemma?

-l521: I think there is a hat missing on the tau
Summary: This paper provides a theoretical work on the asymptotic properties of an estimator of the MSE (as well as an estimator of the noise level) of the Lasso procedure in high dimension.
Though, the theory seem appealing, the practical benefit of the considered method is not conclusive
as is.

Submitted by Assigned_Reviewer_5

(*) The paper proposes new estimators for:
(i.) the risk of a lasso estimator
(ii.) the noise-level in the lasso model.
Both estimators are purely from the data. Under some conditions, it is proven that these are consistent
estimators of the quantities when the dimension p and the number of measurements n go to infinity.
There are also a few experimental results showing the accuracy of the proposed estimators on simulated examples.


(*) The paper provides important results for a fundamental problem of estimating the lasso risk from data, and thus offers potentially
a significant contribution. An important practical implication is that the estimator could be used to optimize the regularization parameter \lambda of the LASSO.
However, the writing of the paper is not clear enough and often sloppy. This makes it hard to follow the paper, and verify
the correctness (and sometimes even the meaning) of the results, and prevented me from assigning a higher score for the paper.
It looks like more effort should have been devoted to presentation.


(*) The computer experiments part is encouraging but limited and not well explained. What is the distribution for the non-zero
values of x_0? how was each point in the plot obtained? from a single run of the lasso? if so, why not average over multiple runs,
to see also the standard deviation of the estimation error and of the estimators the authors provide?


(*) Many things are not explained:
- What are 'irrepresentability conditions' in the introduction?
- How does one initialize y and z in the AMP algorithm in eq. (3.1)?
- I don't understand what is the joint empirical distribution of {\hat{y}_i, x_0,i} (first defined in line 270).
x_0 is the signal, with coordinates for i=1,..,p. But \hat{y} is the the data, with i=1,..,n. Why is the same i used here?
- In line 198 it is said : \eta_t' is the derivative ...'. Derivative with respect to what? y^t?
- What is D in proposition 3.2 (line 297)?
- The authors provide a general estimator for a general design matrix, in definition 3. But when should one use each estimator? is it assumed
that the inverse covariance matrix \Omega is known? or should one estimate it from the data?
- It is often not clear when the authors write Expectations E[..], over what are these expectations taken. For example in equations (3.2), (3.4), (3.5).
Is it over the choice of elements of the matrix A? of error variables Z? of the signal X? all of these together?

(*) The authors provide estimators for the error for one specific estimator: the lasso.
But could it be that other estimators give better MSE for the case of sparse vectors? what would a comparison of the MSE to minimax lower bounds yield?
(e.g. E. J. Candès and M. A. Davenport. How well can we estimate a sparse vector? Applied and Computational Harmonic Analysis, 2011)


(*) I have a problem in interpreting the main results. In theorem (4.1), the authors provide a consistency result for the estimator of the lasso risk,
as both the dimension p and the number of measurements n go together to infinity (with their ratio being fixed). But, first, it looks like the result is given for
a fixed iteration t - yet one would expect the number of iterations required for convergence to depend on n and p (and the data). How do the results apply to the
actual LASSO solution? Second, from eq. (4.1) it looks like n and p approach infinity independently, which is confusing.
Finally, it seems to me that in this limit, the lasso risk itself approaches some limit (i.e. some constant. For example in [Wai09] it is shown that
under certain conditions the error goes to zero). If this is true, then what is the utility of the author's estimator in this limit if the error can be anyway
calculated as a function of the distribution of A and the noise?


(*) There are many typos and inaccuracies, some of which really hinder the reading and understanding:

The last section in the abstract ('We compare our variance ..') is unclear and confusing.
What are 'two naive LASSO estimator'? estimators of the error? also, why 'On the other hand'?

Page 1, line 43: 'p < n' should be 'p > n'?

Page 2, line 69: 'the the risk' -> 'the risk'.

Page 2, line 82: 'data y,A' -> 'data y and A'?

Page 2, line 100: 'underbraccdelta' ???

Page 3, line 132: 'SURE' is written before it was introduced/defined.

Page 3, line 153: 'for each lambda'

Page 3, line 155: \delta is first written here but defined only later.

Page 4, lines 192-196: many indices in eq. (3.1) are wrong/missing (e.g. when is t used, when t+1,t-1? the iteration looks circular). Also, shouldn't the 2nd line be z+Ax?

Page 5, line 260: 'provides' -> 'provide'

Page 6, line 295: 'for' should be removed. Would be also good to define 'weakly differentiable'

Page 9, line 473: Reference is messed up


Summary: The paper provides important results for a fundamental problem of estimating the lasso risk from data, and thus offers potentially
a significant contribution. An important practical implication is that the estimator could be used to optimize the regularization parameter \lambda of the LASSO.
However, the writing of the paper is not clear enough and often sloppy. This makes it hard to follow the paper, and verify
the correctness (and sometimes even the meaning) of the results, and prevented me from assigning a higher score for the paper.
It looks like more effort should have been devoted to presentation.

Submitted by Assigned_Reviewer_6

This is an interesting paper that applies Stein's Unbiased Risk Estimator on the a pseudo data vector, which has a certain distribution shown by previous papers on approximate message passing, to derive an asymptotic estimator of the risk and the noise variance of the lasso.

The results of this paper are significant. The risk formula can inform the choice of the tuning parameter lambda and the variance estimate is certainly useful for residual analysis. The formulas presented are asymptotically correct and the authors show some simulations in which the formulas are accurate with 5000 samples.

The theory used and developed by the paper is not at all straightforward and the paper is well-written given its complexity. I could only understand the outline of the proof and I can only trust that the details are correct.

My only complaint about this paper is the shortage of simulation experiments. It would be very interesting to see how the estimation formulas perform as we vary n,p. I tried implementing the formulas and making my own simulations, but the formulas didn't work, even when I tried to replicate the setting of the paper's simulation. I find that $1-b_n$ is often negative, which renders the other estimated quantities senseless. I'd appreciate it if the authors could either share their code for the simulations (through anonymous URL in the rebuttal possibly) or inspect and debug my R code below.

Overall a good paper; good theoretical contribution with possible practical benefits.

\texttt{
library(glmnet)
n=5000; p=10000; sigma=0.5; lambda=0.2; beta=c(rep(1,0.1*p), rep(0,0.9*p));
X = (1/sqrt(n))*matrix(rnorm(n*p), nrow=n, ncol=p);
noise = sigma*rnorm(n); y = X %*% beta + noise;

glmnet.fit = glmnet(x=X, y=y, lambda=lambda*(1/n)) #glmnet uses average loss
beta_hat = glmnet.fit$beta; relvars = which(abs(beta_hat) > 0.000001)

bn = length(relvars)/n;
pseudo.y = beta_hat + (1/(1-bn)) * t(X) %*% (y- X%*% beta_hat)
theta = lambda/(1-bn)
tau = sqrt(sum((y- X %*% beta_hat)^2))/(sqrt(n)*(1-bn))

Rhat = tau^2 - (2*tau^2/p)*sum( abs(pseudo.y) < theta) + (1/p)*sum( (sapply(abs(pseudo.y), FUN=function (x) {min(x,theta)} )^2))
varhat = tau^2 - Rhat
}
Summary: Good theoretical contribution toward an important problem. The results are possibly practical but not easy o use.
Author Feedback

Author rebuttal: REVIEWER 4:

We acknowledge the presence of typos and thank the reviewer for pointing them to us. We will definitely address them.
Regarding the other comments/questions, we think the following points are important to highlight.

(*) Our results are rigorously proven for the case of standard gaussian design model (Theorems 4.1, 4.2, and Corollary 4.3). Even in this case, they are highly non-trivial. We agree that the treatment of general gaussian design model relies on a conjecture that is supported by earlier work and sophisticated statistical physics arguments such as the replica method. The simulation results strongly suggest that our formulas are accurate.

(*) Thanks for pointing to us other methods that analyze the LASSO-like regression methods with unknown noise.

We will definitely cite them as related work.
However, after reading the aforementioned papers, we found that:

(a) None of them provides an estimator for the LASSO risk.

(b) All of them (except Städler-Bühlmann-van de Geer: [SBG12]) assume sparsity of the original signal x0 whereas in our work this condition is not required. We will definitely compare with [SBG12] and a related method by Sun-Zhang'11 in the next version of the paper. However, we did compare our method with refitted-cross validation (RCV) method of [FGH12] for the noise level estimation as they have general assumptions as ours.


(*) Regarding the convergence of n/p to a fixed ratio \delta we note that since our results hold for any constant \delta in (0,\infty), in practice even if p >> n, one can use our formulas using the actual value of n/p as the limit \delta. Indeed the formulas allow to recover known results (with sharp constants) in the regime p >> n.

----------------------------
REVIEWER 5:
-----------

We thank the reviewer for his/her detailed analysis.

(*) Regarding the extension to other estimation methods:

Theorem 4.1 applies to general AMP algorithms where non-linearities \eta_t can be arbitrary, and hence its results apply to a very broad class of iterative methods. Each such algorithm produces a different estimator than the LASSO. While further generalizations might be possible, AMP algorithms have the property of admitting a fairly explicit characterization.

(*) Regarding interpreting the results:

(a) Theorem 4.1 applies to any fixed number of iterations and does not require convergence of the algorithm. This is of independent interest because one might want to halt the iteration when a certain degree of estimation accuracy is achieved. Theorem 4.2 applies to the actual LASSO solution.
(b) p and n are related by the condition that the sequence must be converging (cf. Definition 1 or lines 220-221 of the paper). In particular, the ratio n/p converges.
(c) Existing results (e.g. Bickel et al 2009) imply that the (\ell_2) risk of LASSO vanishes in probability under restricted eigenvalue or similar conditions. However these upper bounds are quite conservative.
For instance, they would not appear in plots on the left panels of Figures 1 and 2 because the upper bounds are very large (out of the plotted range).
While these upper bounds are very useful to support the use of the LASSO or similar methods, they cannot be used for practical matters such as tuning \lambda, as they only estimate the risk within some large constant.

(*) Thanks for pointing the typos. We will definitely address them.
Below are responses to the reviewer's other questions:

- The distribution of x0 can be arbitrary as long as its empirical distribution converges and the limit has a finite second moment (cf. definition 1).
In the simulations coordinates of x0 are iid: equal to 1 wp .05, -1 wp 0.05, or 0 wp 0.9.
- In simulations we use single run of the LASSO to highlight the statistical error of the estimator.
We also note that each point uses new (independent) x0 and noise which allows to guess statistical fluctuations.
Nevertheless, we agree that showing the average values of several runs with standard errors would be useful and we plan to add them in the longer version of the paper.
- Irrepresentability states that LASSO selects the true model consistently iff the predictors that are not in the true model are "irrepresentable" by predictors that are in the true model (cf. Section 2 of [ZY06] for an exact definition).
- AMP algorithm is initialized by x^0 = y^0 = z^0 = 0.
- Regarding the joint empirical distribution of {\hat{y}_i, x_0,i}, i goes from 1 to p for \hat{y}_i as well since \hat{y} is in R^p.
- In line 198 \eta_t' is derivative with respect to y^t. Since it is assumed to be separable, the derivative is also applied coordinate-wise.
- In line 297, D is the multivariate derivative.
- In practice, the inverse covariance can be estimated from the data.
- All expectations E[] are with respect to the joint distribution of all random variables inside the brackets: i.e., either wrt A, x0, and noise or wrt X and Z.

----------------------------
REVIEWER 6:
-----------

We thank the reviewer for his/her positive comments.

(*) Changing n/p will only change the shape of the curve. The quality of the estimators will remain valid.

(*) We debugged the reviewer's R code and found the following two minor errors:

1) glmnet by default adds an intercept and standardizes the data so these options should be turned off (the latter needs glmnet version 1.93 or beyond).
The correct line in R should be:

glmnet.fit = glmnet(x=X, y=y, lambda=lambda/n, standardize = FALSE, intercept=FALSE)

2) This one is caused by a latex error in line 100 of the paper: Rhat should be divided by delta.
The formula is stated correctly in Corollary 4.3.
Therefore, the correct line in R should be (given that delta=n/p):

varhat = tau^2 - Rhat/(n/p)

The code will run perfectly after these changes (i.e., varhat will be close to sigma^2=0.25).